# Timing is everything: Early do-not-resuscitate orders in the intensive care unit and patient outcomes

Daniel J. Ouyang[1,2,‡], Lindsay Lief[1,2,‡], David Russell[1,3], Jiehui Xu[1], David A. Berlin[1,2], Eliza Gentzler[1], Amanda Su[1], Zara R. Cooper[4], Steven S. Senglaub[4], Paul K. Maciejewski[1,2,5], Holly G. Prigerson[1,2]*

1 Center for Research on End-of-Life Care, Weill Cornell Medicine, New York, New York, United State of America, 2 Department of Medicine, Weill Cornell Medicine, New York, New York, United State of America, 3 Department of Sociology, Appalachian State University, Boone, North Carolina, United States of America, 4 Department of Surgery, Brigham and Women's Hospital, Boston, Massachusetts, United States of America, 5 Department of Radiology, Weill Cornell Medicine, New York, New York, United State of America

‡ These authors are co-first authors on this work.
* hgp2001@med.cornell.edu

**Data Availability Statement:** All relevant data are within the manuscript and its Supporting Information files. Raw data uploaded. Data is stored in secured server held by Weill Cornell

## Abstract

### Background

The use of Do-Not-Resuscitate (DNR) orders has increased but many are placed late in the dying process. This study is to determine the association between the timing of DNR order placement in the intensive care unit (ICU) and nurses' perceptions of patients' distress and quality of death.

### Methods

200 ICU patients and the nurses (n = 83) who took care of them during their last week of life were enrolled from the medical ICU and cardiac care unit of New York Presbyterian Hospital/Weill Cornell Medicine in Manhattan and the surgical ICU at the Brigham and Women's Hospital in Boston. Nurses were interviewed about their perceptions of the patients' quality of death using validated measures. Patients were divided into 3 groups—no DNR, early DNR, late DNR placement during the patient's final ICU stay. Logistic regression analyses modeled perceived patient quality of life as a function of timing of DNR order placement. Patient's comorbidities, length of ICU stay, and procedures were also included in the model.

### Results

59 patients (29.5%) had a DNR placed within 48 hours of ICU admission (early DNR), 110 (55%) placed after 48 hours of ICU admission (late DNR), and 31 (15.5%) had no DNR order placed. Compared to patients without DNR orders, those with an early but not late DNR order placement had significantly fewer non-beneficial procedures and lower odds of being rated by nurses as not being at peace (Adjusted Odds Ratio namely AOR = 0.30; [CI = 0.09–0.94]), and experiencing worst possible death (AOR = 0.31; [CI = 0.1–0.94]) before controlling for procedures; and consistent significance in severe suffering (AOR = 0.34; [CI

Medical School. Access is limited to public due to HIPAA policy on data containing PHI.

**Funding:** The corresponding author Holly Prigerson, PhD receives an R35 fund from NIH. She is the PI for the clinical trial described in the manuscript.

**Competing interests:** The authors have declared that no competing interests exist.

= 0.12–0.96]), and experiencing a severe loss of dignity (AOR = 0.33; [CI = 0.12–0.94]), controlling for non-beneficial procedures.

## Conclusions

Placement of DNR orders within the first 48 hours of the terminal ICU admission was associated with fewer non-beneficial procedures and less perceived suffering and loss of dignity, lower odds of being not at peace and of having the worst possible death.

## Introduction

Cardiopulmonary Resuscitation (CPR) was introduced to clinical practice in the 1960's [1] and became a default treatment for patients with cardiac arrest regardless of their underlying injury or disease.[2] Since that time, however, it has become clear that CPR does not necessarily benefit patients who are terminally ill.[3–5] Do-Not-Resuscitate (DNR) orders are an alternative for patients at the end of life,[3] to prevent receipt of nonbeneficial procedures (e.g. CPR) and unnecessary suffering when patients are imminently dying.[6, 7]

In recent decades, the number of Americans who spend part of their last month of life in the Intensive Care Unit (ICU) has increased to near 30%. Over this same period, the use of DNR orders has increased, [8–11] however most DNR orders are placed very close to the time of death, with a high percentage of DNR orders placed within 24 hours of death.[11–13]

Little is known about the relationship between the timing of DNR orders and patients' quality of death. Results from our recently published report on nurse perception of suffering at the end of life in the ICU did not demonstrate an association between DNR status and quality of death, but did not distinguish early from late DNR.[14] Few studies have examined the timing of DNR orders and its association with mortality, length of stay, interventions, and cost. [15–18] To our knowledge, no previous studies have reported associations between DNR timing and patient-centered outcomes, such as physical or emotional distress, peacefulness, suffering or loss of dignity.

In the present study, we hypothesized that compared to late DNR (orders placed after the first 48 hours of ICU admission), early DNR (orders placed prior to or within the first 48 hours of ICU admission) would be associated with higher quality of death in the ICU; including less nurse-perceived physical distress, psychological distress, suffering and loss of dignity.

## Materials and methods

Weill Cornell Medicine (WCM) Institutional Review Board (IRB) approved this clinical observational trial (IRB 1504016102). IRB approval was obtained from all participating study sites. A full waiver for consent from deceased patients was approved by IRB at New York Presbyterian Hospital/Weill Cornell Medicine (NYP/WCM) and Brigham and Women's Hospital (BWH). Written informed consent was obtained from all nurses participated in the study. All study involving human participants were in accordance with the ethical standards of the institutional and/or national research committee and with the 1964 Helsinki declaration and its later amendments or comparable ethical standards. The study was discussed regularly between the study principal investigator and co-investigators and reviewed by IRB at least once a year to ensure the protocol was rigorously followed.

## Study design

From September 2015 to March 2017, data were collected from nurses to assess the quality of life of 200 patients who died in the Medical ICU (MICU) or Cardiac Care Unit (CCU) of NYP/WCM in Manhattan or the Surgical ICU (SICU) at BWH in Boston. Nurses' evaluations of the quality of life in the patient's last week were assessed. Data from the patients' medical charts were abstracted to confirm clinical information about patients, medical care received and timing of DNR orders.

Each week, trained study staff screened consecutive patients who died in the MICU and CCU at New York Presbyterian Hospital/Weill Cornell Medicine (n = 358), or in the SICU at Brigham and Women's Hospital (n = 64) to identify a nurse who cared for the decedents for at least one 12-hour shift in their last week of life. After obtaining their informed consent, nurses were interviewed individually and in person. Nurse participation occurring outside of work hours was compensated with a $20 gift card per person. Nurses were selected to be the primary assessors of patients' experiences just prior to death because several studies have demonstrated that nurses provide accurate assessments of patients' experience at the end of life, and can accurately predict in-hospital outcomes, particularly when compared to physicians and family members [19–23].

## Data collection

Ninety-eight percent of the nurses approached (100/102) agreed to participate in the study, and 83% (83/100) were selected for data analysis based on the number of shifts, the time between their shifts and patients' death, and presence at patients' death. For some patients, multiple nurses who cared for them were interviewed. And in these cases, we selected the nurses with the most shifts caring for the patient in the last week of life. A variable that captured the time between nurses' last shift and patients' death was used to determine the nurses for analysis if more than one nurse had the same number of shifts; In that case, we selected the nurse whose shift was closest to the time of the patient's death. Nurses who were present at the patient's death were prioritized and selected in this way. Each patient was cared by the interviewed nurse for 2.41 shifts (standard deviation = 1.04), 65 patients had interviewed nurses who were present at their death, and 37 patients had unknown information on if the interviewed nurses were present at the patient's death).

The most common reason that otherwise eligible patients were excluded was due to nurse scheduling conflicts; 70 patients were excluded because they were in the ICU for less than 24 hours and did not have a nurse who took care of them for an entire shift. Trained staff conducted structured clinical interviews with the nurse within three weeks of the patient's death.

Patient demographics, diagnoses, care received, and DNR status were abstracted from medical charts. Orders, admission notes, resuscitation records and death notes were reviewed and checked by trained staff to obtain accurate time and date of ICU admission, DNR placement, and death. Use of life-sustaining therapies, including mechanical ventilation, renal replacement therapy, feeding tubes, and vasopressors was also documented from the medical charts.

## Measures

All of the measures below have been validated in prior published work.[14]

**DNR order status.** The information about DNR orders was collected via inpatient electronic medical record systems. Date and time of each DNR was documented if multiple orders were placed. The person who agreed to sign the DNR and his/her relationship to patient was documented in medical notes. Patients who had a DNR order placed prior to or during the first 48 hours of ICU admission, as documented in the patient's medical chart, were coded as

'Early DNR'. Patients who had a DNR order written after 48 hours of ICU admission were coded as 'Late DNR'. Those who died without a DNR order in place were coded as 'No DNR'.

**Medical care in the last week of life.**   Use of invasive therapies including chemotherapy, vasopressors, dialysis, mechanical ventilation, feeding tubes, cardiac resuscitation, and surgery were abstracted from the medical chart together with other information such as comorbidities, date/time and cause of death. Data were entered into safe, and secure online database. The decision to withdraw life- support was also documented.

**Patient symptoms.**   Nurses evaluated common ICU patient symptoms that may have contributed to suffering. These symptoms included trouble breathing, edema, physical pain, painful broken skin, thirst, nausea or vomiting, fecal incontinence, constipation or diarrhea, urinary incontinence, loss of control of limbs, fever or chills, fatigue and difficulty sleeping.

**Perceptions of patient quality of life and suffering.**   Measures of the patient experience in the last week of life were developed based on prior literature and discussions with ICU physicians, nurses, and end-of-life specialists and validated in a prior study.[14]

During structured interviews, nurses were asked to rate items on a scale from 1 to 10, where 1 was defined as best possible and 10 was defined as worst possible and the items included the decedents' physical and psychological distress, appearing at peace, having the worst possible death, suffering and loss of dignity. The assessment was based on the previously validated questions on patient quality of life in the last week of life.[19] Scores of 8 or higher on this scale were distinguished from lower scores to represent patients with severe symptoms. The suffering and loss of dignity measures were associated with previously validated measures of psychological distress, physical distress, and overall quality of death, and peacefulness at the end of life, [24–27] with results demonstrated highly significant associations (all $p < 0.001$).

## Data analysis

Means and percentages were used to summarize patient characteristics for the total analytic sample and by DNR status. Analysis of variance (ANOVA) or its non-parametric counterpart Kruskal-Wallis test was employed to compare patient characteristics represented by continuous variables, depending on whether the assumption of normal distribution was satisfied. Chi-square tests were used for categorical variables respectively, to test marginal associations between patient characteristics and DNR status. Items with significantly small p-values (<0.05) indicating imbalanced distribution among 3 DNR groups: early, late and no DNR) were considered as possible confounders including age, gender, length of ICU stay and Charlson Comorbidity Index (CCI) and these were adjusted for in the following analyses. Three sets of logistic regression models were then estimated: 1) invasive procedures were regressed on patients' DNR status adjusting for identified possible confounders; 2) nurse-evaluated patient quality of death was regressed on DNR status adjusting for age, gender, race, and length of ICU stay; and 3) nurse-evaluated patient quality of death outcomes were regressed on DNR status, adjusting for same set of confounders, or adjusting for these same confounders and a sum of significant non-beneficial procedures detected above. A p-value of 0.05 was used in all analyses as the threshold for determining statistical significance. R version 3.5.1 was used to perform all statistical analyses.

## Results

Of the 200 assessed decedent patients, 30 (15%) died in the SICU, 25 (12.5%) in the CCU, and 145 (72.5%) in the MICU. 59 patients (29.5%) had a DNR placed within 48 hours of ICU admission (early DNR), 110 (55%) had DNR orders placed after 48 hours of ICU admission (late DNR), and 31 (15.5%) had no DNR order in place.

Most patients were 65 years or older at the time of ICU admission (Median M = 66.9 years; SD = 15.2), male (61.0%), and white (63.5%). Most patients received life-sustaining medical interventions during their ICU stay, the most common were vasopressors (86.5%) and mechanical ventilation (81.5%). DNR orders were placed for 34 patients (17%) after having a cardiac arrest and receiving CPR and 44 patients (22%) received CPR within 48 hours of death.

Patients of different DNR groups varied by age (p<0.001), Charlson Comorbidity Index (CCI) scores (p = 0.025), gender (p = 0.031) and length of ICU stay (p<0.001). The mean age of early DNR patients (M = 73.3) was higher than that of late (M = 66.1) or no DNR (M = 57.5) patients, and their mean CCI score (M = 5.76) greater than late (M = 5.65) or no DNR (M = 4.35) patients, suggesting that older patients, and those with more comorbidities tend to have DNR orders placed early. Although when comparing the difference in decision makers by early, late and no DNR order groups, the p value did not achieve a level of statistical significance p = (0.053), there was a trend suggesting differences in timing of DNR placement by decision-maker. Specifically examining the relationship between decision-maker and each time period we found that DNR orders were more likely to be placed early when decided by patients themselves (Odds Ratio or OR = 2.9, p = 0.039) and were less likely if the spouse made the decision (p<0.01). (Table 1)

After adjusting for possible confounders, patients with an early DNR order in place, compared to those with no DNR, were significantly less likely to receive certain medical interventions during their ICU stay, including dialysis (Adjusted Odds Raito namely AOR = 0.22; [CI = 0.07–0.69]); mechanical ventilation (AOR = 0.16; [CI = 0.03–0.8]); feeding tube (AOR = 0.33; [CI = 0.11–0.96]); cardiac resuscitation (AOR = 0.05; [CI = 0.01–0.2]). No differences were detected between those with a late DNR or no DNR except for cardiac resuscitation (AOR = 0.04; [CI = 0.01–0.12]) and withdraw life support between (AOR = 3.98; [CI = 1.09–14.57]). (Table 2)

Adjusted analyses revealed further that patients with early DNR order placement had lower odds than those with no DNR orders of ratings by nurses indicating poor end-of-life outcomes, including not being at peace (AOR = 0.30; [CI = 0.09–0.94]), experiencing worst possible death (AOR = 0.31; [CI = 0.1–0.94]), suffering (AOR = 0.38; [CI = 0.14–0.99]), and experiencing a loss of dignity (AOR = 0.26; [CI = 0.09–0.7]). However, no difference was detected for the above terms when comparing late DNR patient group to no DNR group. Adjusted odds ratio comparing early vs no DNR group became insignificant for not being at peace (AOR = 0.41 [CI = 0.12–1.38]) and worst possible death (AOR = 0.32; [CI = 0.1–1.02]) when controlling for number of significant invasive procedures, suggesting these procedures accounted for the association between early DNR order placement and those outcomes. (Tables 3–5).

## Discussion

Our results suggest that early DNR order placement (within 48 hours of ICU admission) for patients who die in the ICU is associated with fewer life-sustaining interventions and less nurse-perceived suffering and loss of dignity. Early DNR was also associated with decreased odds of being perceived by nurses as not at peace or having the worst possible death before adjusting for procedures such as dialysis, mechanical ventilation, feeding tube, cardiac resuscitation and withdrawal of life support.

Previous studies have examined the impact of early DNR in ICU patients on cost, procedures and mortality, but this is the first study, to our knowledge, to examine the relationship of DNR timing on patient distress, peacefulness and dignity. Consistent with the published

**Table 1. Patient characteristics and their associations with patients' DNR order status (N = 200).**

| | | DNR Order Status | | | | | | | | ANOVA/CHISQ |
| | | Full Sample | | Early DNR | | Late DNR | | No DNR | | |
| | | N = 200 | | N = 59 (29.5%) | | N = 110 (55.0%) | | N = 31 (15.5%) | | |
| Variable | | mean | SD | mean | SD | mean | SD | mean | SD | p |
| Age (years) | | 66.9 | 15.2 | 73.3 | 13 | 66.1 | 14.2 | 57.5 | 17.1 | <0.001 |
| Charlson Comorbidity Index | | 5.48 | 2.54 | 5.76 | 2.15 | 5.65 | 2.59 | 4.35 | 2.81 | 0.025 |
| Variable | | n | % | n | % | n | % | n | % | p |
| Sex | | | | | | | | | | 0.031 |
| | Male | 122 | 61.0% | 28 | 47.5% | 75 | 68.2% | 19 | 61.3% | |
| | Female | 78 | 39.0% | 31 | 52.5% | 35 | 31.8% | 12 | 38.7% | |
| Race | | | | | | | | | | 0.052 |
| | White | 127 | 63.5% | 43 | 72.9% | 69 | 62.7% | 15 | 48.4% | |
| | Non-White | 68 | 34.0% | 15 | 25.4% | 37 | 33.6% | 16 | 51.6% | |
| DNR by | | | | | | | | | | 0.053 |
| | Spouse | 67 | 39.6% | 17 | 28.8% | 50 | 45.5% | -- | -- | |
| | Family | 67 | 39.6% | 24 | 40.7% | 43 | 39.1% | -- | -- | |
| | Non-Family | 16 | 9.5% | 7 | 11.9% | 9 | 8.2% | -- | -- | |
| | Patient | 19 | 11.2% | 11 | 18.6% | 8 | 7.3% | -- | -- | |
| Diagnosis | | | | | | | | | | 0.175 |
| | Respiratory Failure | 63 | 31.5% | 16 | 27.1% | 36 | 32.7% | 11 | 35.5% | |
| | Cardiac arrest | 23 | 11.5% | 9 | 15.3% | 8 | 7.3% | 6 | 19.4% | |
| | Sepsis/Septic Shock | 26 | 13.0% | 9 | 15.3% | 14 | 12.7% | 3 | 9.7% | |
| | Hemorrhage | 21 | 10.5% | 8 | 13.6% | 8 | 7.3% | 5 | 16.1% | |
| | Other | 67 | 33.5% | 17 | 28.8% | 44 | 40.0% | 6 | 19.4% | |
| Variable | | M | IQR | M | IQR | M | IQR | M | IQR | p |
| Days in ICU | | 7 | 8.25 | 3 | 3 | 9 | 10 | 5 | 7 | <0.001* |

**Notes:** M = Median; IQR = Inter Quantile Range; SD = Standard Deviation

*p-value of Kruskal-Wallis test.

**Table 2. Patient care in the last week of life and its associations with patients' DNR order status (N = 200).**

| | DNR Order Status | | | | | | | | | | | | |
| | Full Sample | | Early DNR | | Late DNR | | No DNR | | | | | | |
| | N = 200 | | N = 59 | | N = 110 | | N = 31 | | Early vs No | | | Late vs No | | |
| Procedure | n | % | n | % | n | % | n | % | AOR[1] | CI | p | AOR[1] | CI | p |
| Chemotherapy | 20 | 10.0% | 3 | 5.1% | 13 | 11.8% | 4 | 12.9% | 0.60 | (0.11,3.33) | 0.56 | 1.30 | (0.36,4.73) | 0.69 |
| Vasopressors | 173 | 86.5% | 50 | 84.7% | 94 | 85.5% | 29 | 93.5% | 0.66 | (0.12,3.56) | 0.63 | 0.58 | (0.12,2.76) | 0.49 |
| Dialysis | 67 | 33.5% | 7 | 11.9% | 47 | 42.7% | 13 | 41.9% | **0.22** | **(0.07,0.69)** | **0.01** | 1.12 | (0.48,2.61) | 0.79 |
| Mechanical Ventilation | 163 | 81.5% | 39 | 66.1% | 95 | 86.4% | 29 | 93.5% | **0.16** | **(0.03,0.8)** | **0.03** | 0.53 | (0.11,2.52) | 0.42 |
| Feeding Tube | 129 | 64.5% | 28 | 47.5% | 77 | 70.0% | 24 | 77.4% | **0.33** | **(0.11,0.96)** | **0.04** | 0.64 | (0.24,1.74) | 0.38 |
| Cardiac Resuscitation | 65 | 32.5% | 16 | 27.1% | 22 | 20.0% | 27 | 87.1% | **0.05** | **(0.01,0.2)** | **<0.01** | **0.04** | **(0.01,0.12)** | **<0.01** |
| Surgery | 27 | 13.5% | 5 | 8.5% | 17 | 15.5% | 5 | 16.1% | 0.51 | (0.12,2.14) | 0.36 | 0.96 | (0.31,2.97) | 0.94 |
| Withdraw Life Support | 52 | 26.0% | 18 | 30.5% | 31 | 28.2% | 3 | 9.7% | 3.18 | (0.81,12.58) | 0.10 | 3.98 | **(1.09,14.57)** | **0.04** |

**Notes:** AOR = Adjusted Odds Ratio; Associations of DNR status with patient symptoms are adjusted for age, gender, Charlson Comorbidity Index, and length of ICU stay.

**Table 3. Patient quality of life, suffering and their associations with patients' DNR order status (adjusted for different variables).**

| | | | | | DNR Order Status | | | | | | | | | |
|---|---|---|---|---|---|---|---|---|---|---|---|---|---|---|
| | Full Sample | | Early DNR | | Late DNR | | No DNR | | Early vs No | | | Late vs No | | |
| Quality of Death | N (n) | % | N (n) | % | N (n) | % | N (n) | % | OR | CI | p | OR | CI | p |
| Physical Distress | 183(53) | 28.96% | 57(22) | 38.60% | 98(26) | 26.53% | 28(5) | 17.86% | 2.89 | (0.96,8.72) | 0.060 | 1.66 | (0.57,4.82) | 0.351 |
| Psychological Distress | 137(35) | 25.55% | 41(12) | 29.27% | 78(17) | 21.79% | 18(6) | 33.33% | 0.83 | (0.25,2.72) | 0.755 | 0.56 | (0.18,1.7) | 0.305 |
| Not at Peace | 173(45) | 26.01% | 48(8) | 16.67% | 99(26) | 26.26% | 26(11) | 42.31% | **0.27** | **(0.09,0.81)** | **0.019** | 0.49 | (0.2,1.19) | 0.115 |
| Worst Possible Death | 190(67) | 35.26% | 56(24) | 42.86% | 104(37) | 35.58% | 30(6) | 20.00% | **0.33** | **(0.12,0.94)** | **0.038** | 0.45 | (0.17,1.21) | 0.113 |
| Suffering | 199(91) | 45.73% | 58(21) | 36.21% | 110(51) | 46.36% | 31(19) | 61.29% | **0.36** | **(0.15,0.88)** | **0.025** | 0.55 | (0.24,1.23) | 0.145 |
| Loss of Dignity | 194(81) | 41.75% | 55(15) | 27.27% | 108(48) | 44.44% | 31(18) | 58.06% | **0.27** | **(0.11,0.69)** | **0.006** | 0.58 | (0.26,1.3) | 0.183 |

Notes: OR = Odds Ratio; Bivariate associations of DNR status with patient symptoms.

literature on the subject, [11–13] most patients in this cohort had a late DNR. As others have published, older patients, those with more comorbidities, and those who were white were more likely to have an early DNR [4, 28–30] Older patients and those with comorbidities may have more opportunity for discussion with their doctors and families about advanced directives and may be more likely to have accepted their own mortality.[31, 32] The racial disparity may be, in part, due to distrust of the health care system among patients who are members of racial or ethnic minority groups [33–35] who may perceive DNR orders as denying patients life-saving medical care.

These results highlight the difficulty family surrogates have in making decisions for their loved ones at the end of life while patients themselves are more likely to decide on early DNR. Several studies have shown the psychological stress placed on loved ones making decisions in the ICU, and these stresses persist after the loved one's death. Having conversations about DNR before or early in the ICU stay, when patients are more likely to have the capacity to make their own decisions not only promotes patient autonomy but also a higher probability of receipt of care concordant with their wishes. It may also save the family the additional stress of making these difficult decisions.[36, 37]

Invasive procedures at the end of life have been associated with poor quality of death.[38, 39] Patients who complete advance directives, including DNR orders, are less likely to receive nonbeneficial aggressive care at the end of life[40] and more likely to receive care consistent with their preferences.[41] Although the DNR order itself does not directly impact care until the moment of cardiac arrest, we found that decedents with early DNR received fewer invasive

**Table 4. Patient quality of life, suffering and their associations with patients' DNR order status (adjusted for different variables).**

| | | | | | DNR Order Status | | | | | | | | | |
|---|---|---|---|---|---|---|---|---|---|---|---|---|---|---|
| | Full Sample | | Early DNR | | Late DNR | | No DNR | | Early vs No | | | Late vs No | | |
| Quality of Death | N (n) | % | N (n) | % | N (n) | % | N (n) | % | AOR | CI | p | AOR | CI | p |
| Physical Distress | 183(53) | 28.96% | 57(22) | 38.60% | 98(26) | 26.53% | 28(5) | 17.86% | 2.65 | (0.82,8.51) | 0.103 | 1.61 | (0.55,4.74) | 0.386 |
| Psychological Distress | 137(35) | 25.55% | 41(12) | 29.27% | 78(17) | 21.79% | 18(6) | 33.33% | 0.79 | (0.22,2.89) | 0.727 | 0.51 | (0.16,1.62) | 0.255 |
| Not at Peace | 173(45) | 26.01% | 48(8) | 16.67% | 99(26) | 26.26% | 26(11) | 42.31% | **0.30** | **(0.09,0.94)** | **0.038** | 0.47 | (0.19,1.2) | 0.115 |
| Worst Possible Death | 190(67) | 35.26% | 56(24) | 42.86% | 104(37) | 35.58% | 30(6) | 20.00% | **0.31** | **(0.1,0.94)** | **0.039** | 0.42 | (0.15,1.14) | 0.090 |
| Suffering | 199(91) | 45.73% | 58(21) | 36.21% | 110(51) | 46.36% | 31(19) | 61.29% | **0.38** | **(0.14,0.99)** | **0.048** | 0.55 | (0.24,1.28) | 0.165 |
| Loss of Dignity | 194(81) | 41.75% | 55(15) | 27.27% | 108(48) | 44.44% | 31(18) | 58.06% | **0.26** | **(0.09,0.7)** | **0.008** | 0.59 | (0.26,1.35) | 0.212 |

Notes: AOR = Adjusted Odds Ratio; Associations of DNR status with patient symptoms are adjusted for age, gender, CCI, length of ICU stay.

Table 5. Patient quality of life, suffering and their associations with patients' DNR order status (adjusted for different variables).

| Quality of death outcomes | Full Sample | | Early DNR | | Late DNR | | No DNR | | Early vs No | | | Late vs No | | |
|---|---|---|---|---|---|---|---|---|---|---|---|---|---|---|
| | N (n) | % | N (n) | % | N (n) | % | N (n) | % | AOR | CI | p | AOR | CI | p |
| Physical Distress | 183(53) | 28.96% | 57(22) | 38.60% | 98(26) | 26.53% | 28(5) | 17.86% | 3.26 | (0.96,11.1) | 0.059 | 1.78 | (0.6,5.32) | 0.301 |
| Psychological Distress | 137(35) | 25.55% | 41(12) | 29.27% | 78(17) | 21.79% | 18(6) | 33.33% | 1.03 | (0.26,4.02) | 0.964 | 0.57 | (0.18,1.83) | 0.345 |
| Not at Peace | 173(45) | 26.01% | 48(8) | 16.67% | 99(26) | 26.26% | 26(11) | 42.31% | 0.41 | (0.12,1.38) | 0.150 | 0.56 | (0.22,1.45) | 0.234 |
| Worst Possible Death | 190(67) | 35.26% | 56(24) | 42.86% | 104(37) | 35.58% | 30(6) | 20.00% | 0.32 | (0.1,1.02) | 0.054 | 0.42 | (0.15,1.17) | 0.097 |
| Suffering | 199(91) | 45.73% | 58(21) | 36.21% | 110(51) | 46.36% | 31(19) | 61.29% | **0.34** | **(0.12,0.96)** | **0.041** | 0.53 | (0.23,4.35) | 0.142 |
| Loss of Dignity | 194(81) | 41.75% | 55(15) | 27.27% | 108(48) | 44.44% | 31(18) | 58.06% | **0.33** | **(0.12,0.94)** | **0.038** | 0.66 | (0.29,1.54) | 0.340 |

**Notes:** AOR = Adjusted Odds Ratio; Associations of DNR status with patient symptoms are adjusted for age, gender, CCI, length of ICU stay, and number of procedures taken among dialysis, mechanical ventilation, feeding tube, cardiac resuscitation, and withdraw life support.

interventions in the last week of life, including dialysis, mechanical ventilation, feeding tubes, and CPR compared to those with late DNR and no DNR. These associations may be explained by early conversations about goals of care including invasive procedures, such as dialysis, in addition to DNR orders. Along similar lines, early DNR orders may have been placed along with orders for comfort-focused care, which generally does not include invasive procedures. Understanding the nature and breadth of the conversations which led to the DNR orders is beyond the scope of this study.

All interviewed nurses were blinded on the topic of possible analysis between the timing of DNR and quality of death, and compared to patients with no DNR order, those patients with early DNR orders had significantly lower odds of being not at peace, having the worst possible death, suffering or loss of dignity even after adjusting for confounders including age, gender, CCI and length of ICU stay. Further adjustment for invasive procedures explained away the association between early DNR and peacefulness and having the worst possible death. This suggests that invasive procedures may be the mechanism by which prolonging the dying process is associated with less peacefulness and the worst possible death. Alternatively, nurses may be more comfortable giving opioids for pain or providing anxiolysis with a DNR order in place, therefore an early DNR order may allow for improved symptom management at the end of life.

This is the first study, to our knowledge, to investigate the relationship between DNR timing and ICU patients' physical and psychological suffering, though the results must be examined in light of its strengths and weaknesses. Strengths include the multi-centered sampling and high rate of nurse participation, which increases the study's generalizability and limits selection bias, respectively. Weaknesses include retrospective evaluation of nurses' assessments of patient experience in the last week of life. Because this study interviewed nurses in the weeks after a patient for whom they cared had died in the ICU, recall bias may have affected nurses' ability to rate patient symptoms and suffering. Still, we have no reason to believe that recall bias would influence nurse perception of suffering as it relates to the decedents' DNR status. Another limitation of this study is the nurse-assessment of patient symptoms. While patients' own reporting of their symptoms would be preferable to nurse report, this approach was not feasible due to the observation that a majority of dying patients in the ICU are unable to communicate.[14] Further, earlier studies have demonstrated that nurses provide accurate

assessments of patients' symptoms and in-hospital outcomes at the end of life, especially compared to caregivers and physicians.[14, 19–23, 42] As noted above, this study included a sample of decedents in the ICU, but we recognize that in clinical practice it may be difficulty to know precisely when patients will die. In patients with end-stage disease (e.g., advanced cancer) who have a high predicted mortality, our results suggest that an early approach to conversations about DNR status may reduce avoidable suffering.

In conclusion, early DNR, within the first 48 hours of ICU admission, for patients who die in the ICU is associated with fewer nonbeneficial procedures and lower odds of nurse-perceived loss of dignity, being not at peace, suffering and having had the worst possible death. The timing, not just the presence, of DNR orders may play an important role in patients' quality of death in the ICU.

## Supporting information

**S1 File. DNR data.csv.**
(CSV)

**S2 File. Quality of Life in the last week of life in the ICU _ REDCap.**
(PDF)

## Author Contributions

**Conceptualization:** Daniel J. Ouyang, Lindsay Lief, David A. Berlin, Eliza Gentzler, Amanda Su, Steven S. Senglaub, Holly G. Prigerson.

**Data curation:** Paul K. Maciejewski.

**Formal analysis:** David Russell, Jiehui Xu, Paul K. Maciejewski.

**Funding acquisition:** Holly G. Prigerson.

**Investigation:** Holly G. Prigerson.

**Methodology:** Steven S. Senglaub, Paul K. Maciejewski.

**Project administration:** Daniel J. Ouyang, Amanda Su.

**Supervision:** David A. Berlin, Zara R. Cooper, Holly G. Prigerson.

**Validation:** Zara R. Cooper, Holly G. Prigerson.

**Writing – original draft:** Daniel J. Ouyang, Lindsay Lief, David Russell.

**Writing – review & editing:** Daniel J. Ouyang, Lindsay Lief, David Russell, Jiehui Xu, David A. Berlin, Eliza Gentzler, Zara R. Cooper, Steven S. Senglaub, Holly G. Prigerson.

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
