## [Decision Letter · Decision Letter 0]

29 Aug 2019

PONE-D-19-19046

Timing is Everything: Early DNR in the ICU and Patient Outcomes

PLOS ONE

Dear Dr. Ouyang,

Thank you for submitting your manuscript to PLOS ONE. After careful consideration, we feel that it has merit but does not fully meet PLOS ONE’s publication criteria as it currently stands. Therefore, we invite you to submit a revised version of the manuscript that addresses the points raised during the review process.

We would appreciate receiving your revised manuscript by Oct 13 2019 11:59PM. To enhance the reproducibility of your results, we recommend that if applicable you deposit your laboratory protocols in protocols.io, where a protocol can be assigned its own identifier (DOI) such that it can be cited independently in the future. For instructions see: http://journals.plos.org/plosone/s/submission-guidelines#loc-laboratory-protocols

We look forward to receiving your revised manuscript.

Kind regards,

Lars-Peter Kamolz, M.D., Ph.D., M.Sc.

Academic Editor

PLOS ONE

Journal Requirements:

Additional Editor Comments (if provided):

Reviewers' comments:

Reviewer's Responses to Questions

**Comments to the Author**

1. Is the manuscript technically sound, and do the data support the conclusions?

Reviewer #1: Yes

Reviewer #2: Partly

2. Has the statistical analysis been performed appropriately and rigorously? 

Reviewer #1: Yes

Reviewer #2: Yes

3. Have the authors made all data underlying the findings in their manuscript fully available?

Reviewer #1: Yes

Reviewer #2: No

4. Is the manuscript presented in an intelligible fashion and written in standard English?

Reviewer #1: Yes

Reviewer #2: Yes

5. Review Comments to the Author

Reviewer #1: Dear authors,

thank you for the opportunity to review the manuscript "Timing is Everything: Early DNR in the ICU and Patient Outcomes". Firstly, I would like to congratulate you for having drafted that very relevant and interesting research, this is, in principle, a well-structured / logically structured study on the basis of an acceptable data pool. However, prior a possible publication, I would like to share my thoughts on the manuscript:

1) Please try and design the abstract with a little bit of background information. Consider dividing the abstract in sections (introduction, methods etc.) for a more legible way.

2) You state that nurses, who cared for the decedents for at least a 12-hour shift in their last week of life, were identified for assessments. However, you do not specify, on which day or how many days prior to the patients death they had their last shift. In my opinion, this is a very important factor, because well-beeing from terminally ill patients can change within a short time and may therefore lead to bias in nurses' assessments. Further, you state that nurses were selected based on their presence at patients' death but it is not clear, if that was definitely an inclusion criteria.

3) In Table 1 you also report "Religion" and "DNR by" information. However, these data were not discussed or mentioned.There is a clear difference between early DNR and late DNR at the "spouse" section and a significant higher percentage of "DNR by patient" in the early DNR group. Please discuss the impact of patients' whishes on the timing of DNR placement.

4) You do not specify the nurses' work experience. Assessment of quality of death may therefore be biased due to their individual attitude and experience with terminally ill patients.

Thank you

Reviewer #2: Dear Authors,

Thank you for the opportunity to review the manuscript, “Timing is Everything: Early DNR in the ICU and Patient Outcomes.”

A few remarks:

PlosOne’s required citation style is to be used correctly.

Please avoid abbreviations in the title.

Possibly, the readability of the article would benefit from a “structured” abstract.

More recent publications on the topic (if possible) would be desirable.

Please describe the Measures (p. 5, literature source 12) in detail.

The interview guide is missing (structured interview? Questionnaire?...)

6. PLOS authors have the option to publish the peer review history of their article (what does this mean?). If published, this will include your full peer review and any attached files.

Reviewer #1: No

Reviewer #2: No

---

## [Author Response · Author response to Decision Letter 0]

29 Sep 2019

Please see attach rebuttal letter.

---

## [Decision Letter · Decision Letter 1]

25 Oct 2019

PONE-D-19-19046R1

Timing is Everything: Early Do-Not-Resuscitate Orders in the Intensive Care Unit and Patient Outcomes

PLOS ONE

Dear Dr. Ouyang,

Thank you for submitting your manuscript to PLOS ONE. After careful consideration, we feel that it has merit but does not fully meet PLOS ONE’s publication criteria as it currently stands. Therefore, we invite you to submit a revised version of the manuscript that addresses the points raised during the review process.

We would appreciate receiving your revised manuscript by Dec 09 2019 11:59PM. To enhance the reproducibility of your results, we recommend that if applicable you deposit your laboratory protocols in protocols.io, where a protocol can be assigned its own identifier (DOI) such that it can be cited independently in the future. For instructions see: http://journals.plos.org/plosone/s/submission-guidelines#loc-laboratory-protocols

We look forward to receiving your revised manuscript.

Kind regards,

Lars-Peter Kamolz, M.D., Ph.D., M.Sc.

Academic Editor

PLOS ONE

Reviewers' comments:

Reviewer's Responses to Questions

**Comments to the Author**

1. If the authors have adequately addressed your comments raised in a previous round of review and you feel that this manuscript is now acceptable for publication, you may indicate that here to bypass the “Comments to the Author” section, enter your conflict of interest statement in the “Confidential to Editor” section, and submit your "Accept" recommendation.

Reviewer #1: All comments have been addressed

Reviewer #2: All comments have been addressed

2. Is the manuscript technically sound, and do the data support the conclusions?

Reviewer #1: Yes

Reviewer #2: (No Response)

3. Has the statistical analysis been performed appropriately and rigorously? 

Reviewer #1: Yes

Reviewer #2: (No Response)

4. Have the authors made all data underlying the findings in their manuscript fully available?

Reviewer #1: Yes

Reviewer #2: (No Response)

5. Is the manuscript presented in an intelligible fashion and written in standard English?

Reviewer #1: Yes

Reviewer #2: (No Response)

6. Review Comments to the Author

Reviewer #1: Dear Authors,

thank you for your thorough revision. However, I noticed some more aspects that could again improve the manuscript:

1) Please use new abbrevations consistently and give the written description before using the abbreviations only: end of life vs. EoL vs. EOL.

Within the abtract, AOR is just used as an Abbreviation without explanation.

2)In Table 3a,b and c you marked "worst possible death" with a star, but the explanation under the table is missing.

Other than that, I feel your manuscript has already improved a lot and I am very satisfied with the already performed revision.

Thank you.

Reviewer #2: (No Response)

7. PLOS authors have the option to publish the peer review history of their article (what does this mean?). If published, this will include your full peer review and any attached files.

Reviewer #1: No

Reviewer #2: No

---

## [Author Response · Author response to Decision Letter 1]

16 Dec 2019

Please see attached updated response to reviewers

---

## [Decision Letter · Decision Letter 2]

6 Jan 2020

Timing is Everything: Early Do-Not-Resuscitate Orders in the Intensive Care Unit and Patient Outcomes

PONE-D-19-19046R2

Dear Dr. Ouyang,

We are pleased to inform you that your manuscript has been judged scientifically suitable for publication and will be formally accepted for publication once it complies with all outstanding technical requirements.

With kind regards,

Lars-Peter Kamolz, M.D., Ph.D., M.Sc.

Academic Editor

PLOS ONE

Additional Editor Comments (optional):

Reviewers' comments:

Reviewer's Responses to Questions

**Comments to the Author**

1. If the authors have adequately addressed your comments raised in a previous round of review and you feel that this manuscript is now acceptable for publication, you may indicate that here to bypass the “Comments to the Author” section, enter your conflict of interest statement in the “Confidential to Editor” section, and submit your "Accept" recommendation.

Reviewer #1: All comments have been addressed

Reviewer #2: (No Response)

2. Is the manuscript technically sound, and do the data support the conclusions?

Reviewer #1: Yes

Reviewer #2: (No Response)

3. Has the statistical analysis been performed appropriately and rigorously? 

Reviewer #1: Yes

Reviewer #2: (No Response)

4. Have the authors made all data underlying the findings in their manuscript fully available?

Reviewer #1: Yes

Reviewer #2: (No Response)

5. Is the manuscript presented in an intelligible fashion and written in standard English?

Reviewer #1: Yes

Reviewer #2: (No Response)

6. Review Comments to the Author

Reviewer #1: Dear authors,

Thank you for your revised manuscript in which you adressed every comment. In my opinion, you've drafted a very relevant research topic.

Thank you

Reviewer #2: (No Response)

7. PLOS authors have the option to publish the peer review history of their article (what does this mean?). If published, this will include your full peer review and any attached files.

Reviewer #1: No

Reviewer #2: No

---

## [Editor Report · Acceptance letter]

31 Jan 2020

PONE-D-19-19046R2 

Timing is Everything: Early Do-Not-Resuscitate Orders in the Intensive Care Unit and Patient Outcomes 

Dear Dr. Ouyang:

I am pleased to inform you that your manuscript has been deemed suitable for publication in PLOS ONE. Congratulations! Your manuscript is now with our production department. 

With kind regards,

on behalf of

Dr. Lars-Peter Kamolz 

Academic Editor

PLOS ONE